# SARS-CoV-2 Genome Variations in Viral Shedding of an Immunocompromised Patient with Non-Hodgkin’s Lymphoma

**DOI:** 10.3390/v15020377

**Published:** 2023-01-28

**Authors:** Rodrigo Villaseñor-Echavarri, Laura Gomez-Romero, Alexandra Martin-Onraet, Luis A. Herrera, Marco A. Escobar-Arrazola, Oscar A. Ramirez-Vega, Corazón Barrientos-Flores, Alfredo Mendoza-Vargas, Alfredo Hidalgo-Miranda, Diana Vilar-Compte, Alberto Cedro-Tanda

**Affiliations:** 1Department of Infectious Diseases, Instituto Nacional de Cancerología, Mexico City 14080, Mexico; 2School of Medicine, Universidad Panamericana, Mexico City 03920, Mexico; 3Instituto Nacional de Medicina Genómica, Mexico City 14610, Mexico; 4Unidad de Investigación Biomédica en Cáncer, Instituto Nacional de Cancerología-Instituto de Investigaciones Biomédicas, UNAM, Mexico City 14080, Mexico

**Keywords:** SARS-CoV-2, COVID-19, mutational landscape, non-Hodgkin’s lymphoma

## Abstract

Background. Severe acute respiratory syndrome coronavirus 2 (SARS-CoV-2) causing coronavirus disease 2019 (COVID-19) is the most transmissible ß-coronavirus in history, affecting all population groups. Immunocompromised patients, particularly cancer patients, have been highlighted as a reservoir to promote accumulation of viral mutations throughout persistent infection. Case presentation. We aimed to describe the clinical course and SARS-CoV-2 mutation profile for 102 days in an immunocompromised patient with non-Hodgkin’s lymphoma and COVID-19. We used RT-qPCR to quantify SARS-CoV-2 viral load over time and whole-virus genome sequencing to identify viral lineage and mutation profile. The patient presented with a persistent infection through 102 days while being treated with cytotoxic chemotherapy for non-Hodgkin’s lymphoma and received targeted therapy for COVID-19 with remdesivir and hyperimmune plasma. All sequenced samples belonged to the BA.1.1 lineage. We detected nine amino acid substitutions in five viral genes (Nucleocapsid, ORF1a, ORF1b, ORF13a, and ORF9b), grouped in two clusters: the first cluster with amino acid substitutions only detected on days 39 and 87 of sample collection, and the second cluster with amino acid substitutions only detected on day 95 of sample collection. The Spike gene remained unchanged in all samples. Viral load was dynamic but consistent with the disease flares. Conclusions. This report shows that the multiple mutations that occur in an immunocompromised patient with persistent COVID-19 could provide information regarding viral evolution and emergence of new SARS-CoV-2 variants.

## 1. Introduction

SARS-CoV-2 is the most transmissible ß-coronavirus in history [1]. In comparison with other groups of coronaviruses such as SARS-CoV and MERS-CoV, SARS-CoV-2 acquired significant capacity in terms of speed of spread and ubiquity that led to the increase in COVID-19 positive cases globally, resulting in a health emergency and the decree of a pandemic, with 670 million reported cases and 6.7 million deaths [2].

Genome sequencing, structural identification, and modifications in viral surface proteins (Spike) have been relevant for the design of effective vaccines and therapies [3]. Immunocompromised patients have been highlighted as an important sector of the vulnerable population and a reservoir to promote accumulation of viral mutations throughout the persistent infection lasting from 30 to 168 days of active infection according to several reports [4,5,6,7,8]. 

In this work, we describe the clinical follow-up and viral mutation profile in an immunocompromised patient with non-Hodgkin’s lymphoma over 102 days. The mutational signature found in this SARS-CoV-2 patient was similar to that reported in other publications. We found some biologically relevant mutations.

## 2. Methods

Sample collection. Samples were collected with a flexible nylon swab to reach the nasopharynx. The swab was left in place for several seconds and slowly removed while rotating. The swab was then placed in 2 mL of sterile viral transport medium. 

SARS-CoV-2 RNA extraction and RT-qPCR detection. Total nucleic acid was extracted from 300 µL of viral transport medium from the NPSs or 300 µL of whole saliva using the MagMAX Viral/Pathogen Nucleic Acid Isolation Kit (Thermo Fisher Scientific, Waltham, MA, USA) and eluted into 50 µL of elution buffer. For SARS-CoV-2 RNA detection, 5 µL of RNA template was tested using TaqPath master mix (Thermo Fisher Scientific, Waltham, MA, USA). All tests were run on a Thermo Fisher ABI QuantStudio 5 real-time thermal cycler (Thermo Fisher Scientific, Waltham, MA, USA). 

Illumina Sequencing. The libraries were prepared using the Illumina COVID-seq kit, following the manufacturer’s instructions. First-strand synthesis was carried out with RNA samples. The synthesized cDNA was amplified using ARTIC primers V4, then was tagmented and adapted using IDT for the Illumina Nextera UD Indices Set A, B, C, D (384 indices) (Illumina, San Diego, CA, USA). Dual-indexed pair-end sequencing with a 150 bp read length was carried out on the NextSeq 2000 platform (Illumina, San Diego, CA, USA).

Oxford Nanopore Sequencing. Libraries were prepared according to ARTIC Midnight protocol PCR tiling of SARS-CoV-2 virus with a rapid barcoding kit (SQK-RBK110.96) and sequenced on the GridION sequencing platform. We used the PCRT_9125_v110_revE_24Mar2021 protocol. A total of 800 ng of DNA library was loaded into a primed R.9 flow cell (FLO-MIN106). MinKNOW software v.21.11.7 (Oxford Nanopore Technologies, Oxford, Oxfordshire, UK) was used to collect raw sequencing data. Oxford Nanopore Raw Data Processing and Sequencing Data Quality Assessment Basecalling and barcode demultiplexing were performed with EPI2ME Agent v3.5.4 using the Fastq QC+ARTIC+NextClade app.

## 3. Case Presentation

### 3.1. Clinical Case Description

A 45-year-old patient with a history of non-Hodgkin’s lymphoma presented with recurrent symptomatic SARS-CoV-2 infection. Past medical history was unremarkable except for smoking since 16 years of age. He was referred to our hospital in April 2020 due to chronic cough, dyspnea, unintentional weight loss, and a mediastinal tumor; biopsy revealed a CD20+ diffuse large B-cell lymphoma. At the time of diagnosis, he had a massive left pleural effusion, for which an evacuating paracentesis was performed. He received six cycles of R-CHOP chemotherapy and 18 fractions of radiotherapy, achieving a partial remission. A second line of chemotherapy with R-ICE (rituximab + ifosfamide, carboplatin, and etoposide) was started in February 2021. In June 2021, he showed evidence of left chylothorax. From this moment on, the patient required the use of supplemental oxygen through a nasal cannula. In July 2021, central nervous infiltration was diagnosed, and he started chemotherapy with R-ESHAP (rituximab + etoposide, cytarabine, cisplatinum, and methylprednisolones) and intrathecal methotrexate as salvage therapy. He required chemical pleurodesis in August 2021. The last ESHAP was administered on 17 December 2021.

On 29 January 2022, he was diagnosed with mild COVID-19 with a positive PCR test for SARS-CoV-2. He had not been vaccinated against SARS-CoV-2. At the time of diagnosis, he described mild respiratory symptoms without increased oxygen requirements. He evolved favorably without requiring further interventions. High-dose methotrexate was delayed until full symptomatic recovery and completion of the isolation period, and resumed thereafter. On the fourth day after chemotherapy infusion, he developed a low-grade fever and productive cough. Chest computed tomography (CT) revealed right parahilary condensation and air bronchogram. Antibiotic therapy with piperacillin/tazobactam was given for three days and subsequently switched to levofloxacin on an outpatient basis for five days. Three days later, he sought consultation due to persistent fever and was hospitalized for workup. The chest CT scan reported multifocal pneumonia and he was treated with piperacilin/tazobactam. Due to persistent fever, and new cutaneous lesions, a skin biopsy was performed with histopathology showing evidence of hematologic progression. On 3 March, he started fourth-line chemotherapy with bendamustin and brentuximab. As part of the workup and due to persistent fever, a new PCR test for SARS-CoV-2 was performed on 4 March, being positive. On 6 March, he received three days of remdesivir. On 8 March, one unit of 250 mL of hyperimmune convalescent plasma was infused. Fever resolved and he was discharged on 14 March with supplementary oxygen. His PCR was still positive on 11 March, with viral CTs ORF1a: 26.2, Spike: 0, and Nucleocapsid: 25.9.

He received a second dose of chemotherapy on 1 April 2022, and on 13 April he was readmitted for fever and diarrhea, with no respiratory symptoms or changes in his chest CT. He was started on levofloxacin and piperacillin/tazobactam, with a poor response. On 20 April, his oxygen requirements increased, and chest CT reported new ground-glass opacities. The PCR for SARS-CoV-2 was positive, with viral CTs ORF1a: 15.4, Spike: 0, and Nucleocapsid: 14.2. 

He required non-invasive ventilation and was started on baricitinib and dexamethasone, with clinical improvement, with a new flare (fever and increased oxygen requirements) on 29 April, with a positive PCR with viral CTs ORF1a: 15.4, Spike: 0, and Nucleocapsid: 14.2. A new cycle of remdesivir was initiated for 5 days on 3 May with partial remission of the fever. On 6 May, his viral CTs were ORF1a: 26.8, Spike: 0, and Nucleocapsid: 25.3. On 13 May, the viral CTs were ORF1a: 34.2, Spike: 0, and Nucleocapsid: 32.8. A blood sample was analyzed for SARS-CoV-2-neutralizing antibodies on 13 May, between the first and the second convalescent plasma infusions, and there were no detectable antibodies. We were not able to measure SARS-CoV-2 viremia at any moment. He received a second infusion of hyperimmune convalescent plasma on 16 May. He continued with respiratory distress, but he did not accept invasive mechanical ventilation. Due to his unstable condition, no additional workup to discard other pulmonary conditions such as thromboembolism or opportunistic infections was performed. The patient died on 3 May 2022 (Figure 1).

In relation to viral load, the patient presented two flares, both associated with chemotherapy. For the first flare on 3 March, there was clinical improvement with remdesivir and hyperimmune plasma, but viral load follow-up was not performed. For the second flare on 28 April, there was a decrease in the viral load after a new cycle of 5 days of remdesivir, plasma, and immunomodulatory therapy (Figure 1).

### 3.2. SARS-CoV-2 Mutational Landscape

All samples that underwent whole-virus genome sequencing belonged to clade 21K, lineage BA.1.1. Samples collected at days 1, 39, 87, and 95 had genome coverage >98% and the sample collected at day 102 had 90% coverage; interestingly, we observed dynamic behavior related to the presence and absence of amino acid substitutions through the days of collection in the N (nucleocapsid), ORF1a, ORF1b, ORF13a, and ORF9b genes. All analyses were focused on mutations that had a change in at least one sample collection point, indicating that they were not present at all collection times.

We detected two clusters of amino acid substitutions. Cluster 1 showed a cluster of amino acid substitutions appearing on days 39 and 87, but absent on adjacent days 1, 95, and 102 (branch 1 of the dendrogram), and Cluster 2 was characterized by the presence of several substitutions at day 95 but absent on the other days (branch 2 of the dendrogram) (Figure 2A).

In Cluster 1, we found four amino acid substitutions (N:A119S, ORF1a:P927L, ORF1P1427S, and ORF1b:L82F) detected only on days 39 and 87. The ORF1A:T1638I mutation was not detected on the first day of collection, but was detected on the rest of the days, unlike ORF1b:E1167V, which was only detected on the last day of collection. In Cluster 2, we found the presence of 4 mutations only on day 95 of collection (ORF1a:D4165G, ORF1a:T2124I, ORF9b:T95M, and ORF3a:V112F). We did not find changes in the presence or absence of amino acid substitutions in the spike gene (Appendix A) throughout the study.

We found that genomic profiles for samples collected on days 39 and 87 are nearly identical. Interestingly, the day 1 sample could be grouped either with the day 95 sample or with the day 102 sample, the last collection days. The ambiguous clustering of the sample from the last collection date (day 102) could be due to missing data, i.e., genomic positions that could not be called with high confidence (Figure 2B, shown in grey).

### 3.3. Discussion and Conclusions

In this report, we describe the clinical history and viral evolution of a deeply immunosuppressed patient with refractory Non-Hodgkin’s Lymphoma, who was treated with cytotoxic chemotherapy and rituximab. He received targeted treatment for COVID-19 with two cycles of remdesivir and two hyperimmune plasma but presented persistent replication for 102 days and died. Although the cause of death was attributed to COVID-19, this occurred when SARS-CoV-2 viral replication was low. The patient had pulmonary involvement of the lymphoma, and other complications could not be ruled out. The risk of severe cardiovascular events during the first 90 days of acute COVID-19 infection is increased for all patients, being much higher in severe cases such as the one discussed in this report [9]. We were unable to pursue further investigations on the cause of death, and we cannot rule out a cardiovascular event. Moreover, he had received several cycles of dexamethasone, baricitinib, and chemotherapy, all associated with an increased risk of opportunistic infections. Although baricitinib was not associated with an increased risk of opportunistic infections in the major trials, there were no immunosuppressed patients included and caution is usually recommended in this population [10]. Interestingly, at the time of the patient’s death, viral replication had decreased significantly due to the combination of antivirals, convalescent plasma, and immunomodulatory therapy. These combined therapies in immunosuppressed patients with COVID-19 have been studied and reported with mixed results. Specifically, in patients with hematologic malignancies, there are many case reports and case series suggesting the utility of combining convalescent plasma or monoclonal antibodies, with antivirals [11,12,13].

Multiple cases of SARS-CoV-2 evolution have been reported in patients receiving immunosuppressive therapy [8,14,15,16]. Prolonged viral shedding for up to 100 days has been reported in more than 20 patients with hematologic malignancies such as B-cell lymphomas who are receiving anti-CD20 targeted treatment with rituximab [16]. Anti-CD20 monoclonal antibodies are associated with B-cell depletion and decreased humoral response. Patients infected with SARS-CoV2 fail to develop anti-SARS-CoV2 IgG antibodies and have a poor response to vaccines. These patients have also been described as perfect hosts for viral evolution and emergence of mutations [17]. Some reports describe persistent viral replication and the emergence of clinically important mutations in patients treated with rituximab [17]. This has also been reported in immunocompromised patients infected with SARS-CoV-2 that were treated with steroids, specifically prednisolone [18]. Interestingly, the substitution rate for SARS-CoV-2 in an immunocompromised patient was similar to previous estimates indicating no increase in viral replication rate [19].

In this study, we characterized mutations that were present and absent during the infection. We focused the analysis on mutations that were not found on all collection dates. Some of these mutations appeared and disappeared over time, suggesting the possible emergence and dispersion of new and independent viral quasi-species. Two scenarios are possible from the data observed on this study. One scenario could be the intra-individual emergence of viral quasi-species that fluctuates in frequency over time [20,21,22]. A second scenario could be the introduction of new viral quasi-species by a second and independent infection event which could explain the burst of mutations from day 1 to day 37. Previous studies have shown that reinfections occurred mostly by viruses from different genomic lineages, although reinfection can occasionally occur by viruses from the same genomic lineage [23,24].

The SARS-CoV-2 genome studied in this case had a gain of the ORF1a:T1638I mutation from the second NPS collection (day 39). This mutation was previously observed in a patient with lymphoma in remission treated with rituximab and positive to SARS-CoV-2 for 161 days. At day 30 (second NPS collection), the virus showed a gain of the ORF1a:T1638I mutation, which remained for the next 161 days [25]. Likewise, another study also reported the same mutations in a 70-year-old patient diagnosed with non-Hodgkin’s lymphoma, where, in his sequentially collected SPNs, the mutation was not detected at day 34 (first collection), but showed up on day 54 and remained until day 238 of infection, in a patient previously treated with remdesivir on days 47–51 [26]. It has been reported that this mutation causes a partial loss of CD8+ T-cell response, which could indicate the advantage of its selection during viral evolution in the reported cases and our case report [27].

The mutation N:A119S is not present in variant BA.1.1, but in this report, was detected only on days 39 and 87; this amino acid substitution is present in 99.8% of cases of variant P.2, which circulated in Brazil in 2020 and, together with four other amino acid substitutions, evades neutralizing antibodies; the P.2 variant was classified as a variant of importance by WHO [28].

The mutation ORF1b:P1427 has a deleterious effect on the protein ORF1b, implying changes in the fitness of SARS-CoV-2 variants harboring this mutation. This site may be a potential target for the development of therapeutic agents and deserves further functional studies [29]. For the rest of the mutations present during the viral evolution of this case report, no studies were found in the literature proposing their biological function or their impact in the structure of viral antigens. 

The case reported highlights the complexity of treatment of COVID-19 in patients with hematological neoplasia. The patient described in this report showed the different biological scenarios in an individual with immunosuppression and prolonged viral shedding, and exemplifies a niche where the virus can evolve with the potential of mutation emergence and advanced SARS-CoV-2 fitness. Moreover, the description of the change in viral mutations during infection could provide information to learn more about the viral evolution and emergence of new variants [30]. 

## Figures and Tables

**Figure 1 viruses-15-00377-f001:**
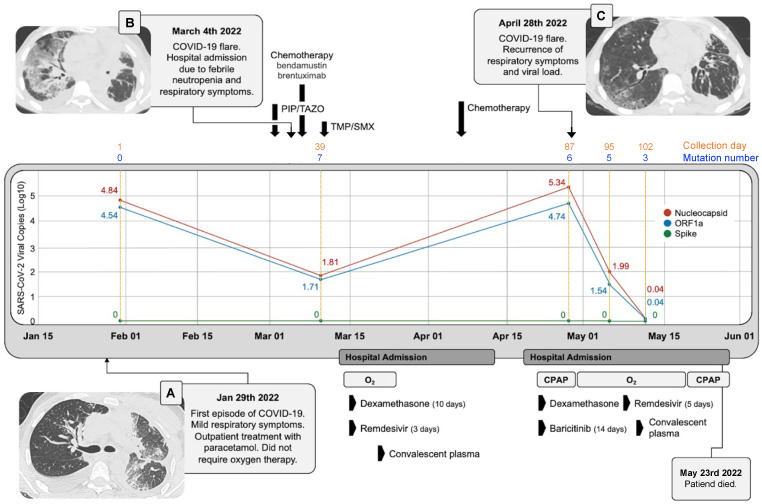
Clinical course and viral load. Timeline of clinical course and SARS-CoV-2 viral load across the five different collection days. The copy number in LOG10 of the three viral genes detected in RT-qPCR is shown. N: Nucleocapsid, S: spike and ORF1a. Solid lines correspond to calendar days and dashed orange lines correspond to elapsed days. The number of mutations per collection date including only those mutations that change their presence/absence status over time is shown per each collection date. Panel (**A**). Chest CT of the first flare. Panel (**B**). Chest CT at COVID-19 diagnosis. Panel (**C**). Chest CT of the second flare.

**Figure 2 viruses-15-00377-f002:**
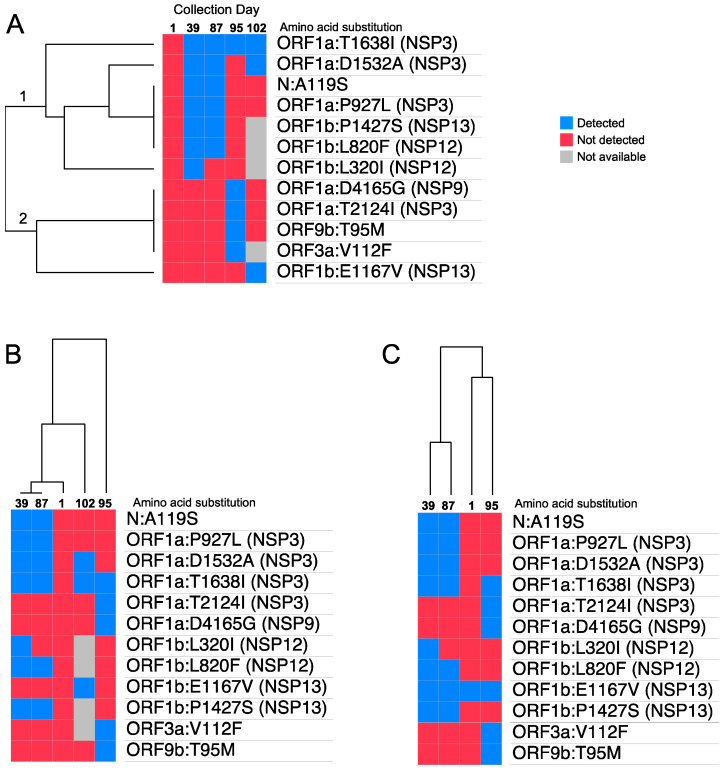
SARS-CoV-2 hierarchical clustering gene expression map. Euclidean distance was used as the distance metric in the hierarchical clustering algorithm. The colored maps show amino acid substitutions with at least one change within days of nasopharyngeal swab collection. Not available means that the region could not be covered during sequencing, so it does not have mutation data. Panel (**A**). Hierarchical clustering was applied over mutations to cluster mutations with similar temporal patterns, and isolates were sorted by collection date. Panel (**B**,**C**). Hierarchical clustering was applied over collection days to cluster isolates with similar mutation profiles, and mutations were sorted by genomic position. In panel (**C**), day 102 was removed from the clustering since 4 out of 12 mutations corresponded to missing data, which diminishes the clustering confidence.

## Data Availability

Genome viral is available in https://gisaid.org/.

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
