# Peer review of "SARS-CoV-2 Genome Variations in Viral Shedding of an Immunocompromised Patient with Non-Hodgkin’s Lymphoma"

_viruses, 2023, doi:10.3390/v15020377_

Round 1

Reviewer 1 Report

First of all, thanks to the authors for submitting this article, in my opinion these cases of Sars-CoV2 infection in immunocompromised patients are interesting and we have to learn about them, but I think the results of this work are a bit superficial and I would like to have a deeper analysis of them.

For example, I would like to know if you find any association between the number of mutations and any of the treatments that the patient received. For example, the use of glucocorticoids, rituximab or bendamustine are very important immunosuppressants and it is possible that their use increased virus replication and the appearance of more mutations.

I would like to know how you consider these mutations. Are they minor mutations, could they have some implication in the fitness or in the structure of the viral antigens?

Another interesting thing is if the SARS-CoV2 viremia could be detected in the patient because this data is more relevant in the balance between viral replication and host immune control and is a prognostic risk factor.

I would like to know if the patient generated antibodies. I know that hyperimmune plasma can interfere with the results, but I would like to have some information about it. I think it is important because of the possibility of using monoclonal antibodies in these patients.

In summary, I think this is an interesting case to publish, but there are many reports about it, and I need more information and a deeper interpretation of these results.

Thank you very much for your work.

Author Response

We thank the reviewers for their valuable comments. We think they improve our manuscript.  Enclosed find a point-by-point response.

Reviewer 1

First of all, thanks to the authors for submitting this article, in my opinion these cases of Sars-CoV2 infection in immunocompromised patients are interesting and we have to learn about them, but I think the results of this work are a bit superficial and I would like to have a deeper analysis of them.

  1. For example, I would like to know if you find any association between the number of mutations and any of the treatments that the patient received. For example, the use of glucocorticoids, rituximab or bendamustine are very important immunosuppressants and it is possible that their use increased virus replication and the appearance of more mutations. 

We thank the reviewer for this comment. As can be seen in Figure 1, the administration of treatments does not correspond with the time of sample collection, besides several treatments were administered at the same time. So, it is difficult to analyze the association between number of mutations and treatment. Moreover, some mutations disappeared over time suggesting the emergence and dispersion of new and independent viral clones. This statement has been added to the manuscript [Lines 172-176]. However, this data is interesting, so we added the number of mutations per collection date in Figure 1.

Regarding the use of immunosuppressants and increased viral replication, we mention in the discussion the role of rituximab in viral persistence [Lines 237-242]. We describe a literature review with more than 20 cases of persistent COVID with the use of  rituximab and other reports describing persistent viral replication as the ground for emerging mutations. We edited the discussion and added more references regarding viral evolution and immunosuppression. Interestingly, the substitution rate for SARS-CoV-2 in a immunocompromised patient was similar to previous estimates indicating no increase in viral replication rate [Lines 243-246].

  1. I would like to know how you consider these mutations. Are they minor mutations, could they have some implication in the fitness or in the structure of the viral antigens? 

The mutations presented in Figure 2 were selected because of their dynamics over time, i.e., these are mutations that had a change in at least one sample collection  implicating that they are not present in all collection times. For example, the ORF1a: T1638I mutation was detected from day 39, but was not present on day 1 of sample collection. Regarding fitness or structure of the viral antigens, only one study was found reporting a protein deleterious effect caused by the ORF1b:P1427 mutation with a potential effect on viral fitness. We found no reports in the literature about any other mutations which could affect  the structure of viral antigens. This has been added in lines 267-271.

  1. Another interesting thing is if the SARS-CoV2 viremia could be detected in the patient because this data is more relevant in the balance between viral replication and host immune control and is a prognostic risk factor. 

Unfortunately we did not measure SARS-COV-2 viremia in this patient.

  1. I would like to know if the patient generated antibodies. I know that hyperimmune plasma can interfere with the results, but I would like to have some information about it. I think it is important because of the possibility of using monoclonal antibodies in these patients. 

We measured neutralizing antibodies on May 13th, after one plasma had been infused, and he did not generate any. We added a sentence in the case description in lines 144-147.

Reviewer 2 Report

There are multiple reports in the literature of immunosuppressed patients with persistent SARS-CoV-2 infections; however, many of these are from early in the pandemic. The current study describes sequential viral genome variation in a patient with non-Hodgkin lymphoma who was chronically infected with an omicron variant. Several sections of the manuscript require clarification. 

  1. The discussion of the mutations in ORF1a and 1b should indicate which non-structural proteins (Nsp’s) are potentially affected by each mutation (for a useful reference, see van de Leemput J, Han Z. Understanding Individual SARS-CoV-2 Proteins for Targeted Drug Development against COVID-19. Mol Cell Biol. 2021 Aug 24;41(9):e0018521. PMID: 34124934). 
  2. The case description in Figure 1 lists specific calendar dates for clinical events, whereas the viral isolates are identified by the number of elapsed days, eg 1, 39, 87, etc. Although the reader can figure out the correlation of days and dates, it would be easier if this information was included in Figure 1. 
  3. Figure 2 is confusing because the dendrogram does not describe the relatedness of the different viral subvariants which is how this type of data is normally presented. The constellation of mutations present in each isolate is contained in the vertical columns (eg day 39, 87, etc), not the rows. The dendrogram should be at the top of the figure and show the relationship between the columns. The horizontal rows in Figure 2 indicate the temporal changes at a specific site, and the existing dendrogram shows the relatedness of these temporal patterns, not the relatedness of the viral isolates. The figure is also confusing because, as a result of the alignments, the rows are not in the linear order of the mutations in the viral genome. 
  4. There are several grammatical errors in the introduction, eg line 42, “631 millions of reported cases” should be “631 million reported cases.” 

Author Response

We thank the reviewers for their valuable comments. We think they improve our manuscript.  Enclosed find a point-by-point response.

Reviewer 2

There are multiple reports in the literature of immunosuppressed patients with persistent SARS-CoV-2 infections; however, many of these are from early in the pandemic. The current study describes sequential viral genome variation in a patient with non-Hodgkin lymphoma who was chronically infected with an omicron variant. Several sections of the manuscript require clarification.

  1. The discussion of the mutations in ORF1a and 1b should indicate which non-structural proteins (Nsp’s) are potentially affected by each mutation (for a useful reference, see van de Leemput J, Han Z. Understanding Individual SARS-CoV-2 Proteins for Targeted Drug Development against COVID-19. Mol Cell Biol. 2021 Aug 24;41(9):e0018521. PMID: 34124934).

We mapped each mutation to the affected Nsps and we included this information in Figure 2.

  1. The case description in Figure 1 lists specific calendar dates for clinical events, whereas the viral isolates are identified by the number of elapsed days, eg 1, 39, 87, etc. Although the reader can figure out the correlation of days and dates, it would be easier if this information was included in Figure 1.

We included the elapsed days in Figure 1 as suggested [Lines 160-165].

  1. Figure 2 is confusing because the dendrogram does not describe the relatedness of the different viral subvariants which is how this type of data is normally presented. The constellation of mutations present in each isolate is contained in the vertical columns (eg day 39, 87, etc), not the rows. The dendrogram should be at the top of the figure and show the relationship between the columns. The horizontal rows in Figure 2 indicate the temporal changes at a specific site, and the existing dendrogram shows the relatedness of these temporal patterns, not the relatedness of the viral isolates. The figure is also confusing because, as a result of the alignments, the rows are not in the linear order of the mutations in the viral genome.

The goal of Figure 2 is to show the temporal changes in the presence and absence of each mutation (each row) highlighting clusters of mutations with similar temporal patterns. Because of this, we applied hierarchical clustering over the mutations and we kept the isolates ordered by date, as a result the mutations are not ordered according to their positions in the genome and the dendrogram shows the relatedness of mutations instead of the relatedness between isolates. We have added a second panel showing the relatedness between isolated and sorting the mutations by genomic position [L183-189]. The genetic similarities are described in lines 199-202.

  1. There are several grammatical errors in the introduction, eg line 42, “631 millions of reported cases” should be “631 million reported cases.”

We have checked the grammar of the introduction very carefully

Round 2

Reviewer 2 Report

The revised manuscript addresses nearly all of the reviewers' comments.

The hierarchical clustering in Fig 2B appears to be inverted. Day 1 is clearly the least related strain and should be on a separate branch, whereas the isolates from days 39 and 87 are nearly identical. The order of similarity appears to be [39=87] > 102 > 95 > 1. Please check this figure. 

Although the manuscript assumes that this patient had a single SARS-CoV-2 strain that evolved over time, I don't think the data excludes the possibility that the patient was infected with a second strain on day 39 which would explain the large number of genetic differences between days 1 and 39. This should be mentioned in the Discussion.

Author Response

The hierarchical clustering in Fig 2B appears to be inverted. Day 1 is clearly the least related strain and should be on a separate branch, whereas the isolates from days 39 and 87 are nearly identical. The order of similarity appears to be [39=87] > 102 > 95 > 1. Please check this figure. 

We double checked the hierarchical clustering algorithm, we were using one-minus pearson correlation which is usually used for continuous gene expression data. However, our mutation data is binary: 0 = not detected, 1 = detected and NA=not available. We analyzed the data again using Euclidean distance since this is better suited for binary data [L187]. Moreover, our confidence about day 102 clustering is low since 4 out of 12 mutations correspond to missing data [L195-196]. Because of this, we present the clustering with and without day 102 [Figure 2]. As the reviewer suggested days 39 and 87 are nearly identical. Interestingly, day 1 could be grouped either with day 95 or with day 102 [207-210].

Response:

Although the manuscript assumes that this patient had a single SARS-CoV-2 strain that evolved over time, I don't think the data excludes the possibility that the patient was infected with a second strain on day 39 which would explain the large number of genetic differences between days 1 and 39. This should be mentioned in the Discussion.

This could be a possible explanation and we have been mentioned it in the Discussion [L254-262]